# National U.S. time-trends in opioid use disorder hospitalizations and associated healthcare utilization and mortality

**Jasvinder A. Singh** [1,2,3]*, **John D. Cleveland** [2]

**1** Department of Medicine, University of Alabama at Birmingham, Birmingham, Alabama, United States of America, **2** Department of Epidemiology, University of Alabama at Birmingham, Birmingham, Alabama, United States of America, **3** Medicine Service, Birmingham VA Medical Center, Birmingham, Alabama, United States of America

* Jasvinder.md@gmail.com

S. time-trends in opioid use disorder
hospitalizations and associated healthcare
utilization and mortality. PLoS ONE 15(2):
e0229174. https://doi.org/10.1371/journal.
pone.0229174

Faculty of Health Sciences, MALTA

**Data Availability Statement:** These data are easily
available from the "Healthcare Cost and Utilization
Project (HCUP)" and can be obtained after
completing an on-line Data Use Agreement training

## Abstract

### Background

The opioid epidemic is a major public health crisis in the U.S. Contemporary data on opioid
use disorder (OUD) related hospitalizations are needed. Our objective was to assess
whether OUD hospitalizations and associated mortality are increasing over time and examine the factors associated healthcare utilization and mortality.

### Methods and findings

We examined the rates of OUD hospitalizations and associated mortality using the U.S.
National Inpatient Sample (NIS) data from 1998–2016. Multivariable-adjusted logistic regression assessed the association of demographic, clinical and hospital characteristics with inpatient mortality and healthcare utilization (total hospital charges, discharge to a rehabilitation
facility, length of hospital stay) during the index hospitalization for opioid use disorder. We
calculated the odds ratio (OR) and 95% confidence intervals (CI). We estimated 781,767
OUD hospitalizations. The rate of OUD hospitalization and associated mortality (/100,000
overall NIS hospitalizations) increased from 59.8 and 1.2 in 1998–2000 to 190.7 and 5.9 in
2015–16, respectively. In the multivariable-adjusted analysis, the following factors were
associated with worse outcomes; compared to age <34 years, older age was associated
with higher risk of hospital charges above the median and length of stay >3 days, slightly
higher risk of discharge to a rehabilitation facility. Higher Deyo-Charlson score was associated with higher hospital charges, length of hospital stay, and inpatient mortality. Women
had lower odds of inpatient mortality than men and blacks had lower odds of mortality than
whites.

### Conclusions

Rising OUD hospitalizations from 1998 to 2016 and increasing associated inpatient mortality
are concerning. Certain groups are at higher risk of poor utilization outcomes and inpatient

session and signing a Data Use Agreement. The contact information for requesting the data is as follows: HCUP Central Distributor Phone: (866) 556-4287 (toll-free) Fax: (866) 792-5313 E-mail: HCUPDistributor@ahrq.gov.

**Funding:** The author(s) received no specific funding for this work.

**Competing interests:** I have read the journal's policy and the authors of this manuscript have the following competing interests. JAS has received consultant fees from Crealta/Horizon, Medisys, Fidia, UBM LLC, Medscape, WebMD, Clinical Care options, Clearview healthcare partners, Putnam associates, Spherix, the National Institutes of Health and the American College of Rheumatology. JAS owns stock options in Amarin pharmaceuticals and Viking therapeutics. JAS is a member of the executive of OMERACT, an organization that develops outcome measures in rheumatology and receives arms-length funding from 36 companies. JAS serves on the FDA Arthritis Advisory Committee. JAS is a member of the Veterans Affairs Rheumatology Field Advisory Committee. JAS is the editor and the Director of the UAB Cochrane Musculoskeletal Group Satellite Center on Network Meta-analysis. JAS previously served as a member of the following committees: member, the American College of Rheumatology's (ACR) Annual Meeting Planning Committee (AMPC) and Quality of Care Committees, the Chair of the ACR Meet-the-Professor, Workshop and Study Group Subcommittee and the co-chair of the ACR Criteria and Response Criteria subcommittee. JDC has no conflicts. This does not alter our adherence to PLOS ONE policies on sharing data and materials.

**Abbreviations:** CI, confidence interval; HR, Hazard ratio; ICD-10-CM, International Classification of Diseases, Tenth Revision, Clinical Modification; ICD-9-CM, International Classification of Diseases, Ninth Revision, Clinical Modification; NIS, National Inpatient Sample; OUD, opioid use disorder; SD, standard deviation; SE, standard error; UAB, University of Alabama at Birmingham.

mortality. Resources and healthcare policies need to focus on the high-risk group to reduce mortality and associated utilization.

## Introduction

The opioid epidemic in the U.S. is a concern for providers, hospitals, policy-makers and the public [1–3]. The opioid epidemic is associated with significant mortality with calls for action to end the epidemic [4,5]. Based on national vital statistics data, the Centers for Disease Control (CDC) reported that 28,647 opioid-related deaths in 2014 that increased to 33,091 in 2015 (16% increase) [1], and to 42,249 deaths in 2016 (47% increase) [6]. The opioid overdose death rate increased from 2000 to 2014 [3] and continued the upward trend, increasing from 9.0 per 100,000 in 2014 to 10.4 in 2015 [1].

In addition to examining opioid use disorder (OUD)-related mortality, hospitalizations associated with OUD can help us better understand the opioid epidemic. A recent study of U. S. national inpatient sample (NIS) documented that nearly half a million hospitalizations yearly included a diagnosis of OUD (in any position, primary or secondary) [7]. Regional and demographic differences exist in prescription opioid and heroin-related overdose hospitalizations [8]. The Centers for Disease Control (CDC) issued a guideline to reduce the overutilization of prescription opioid use as a potential solution to the opioid epidemic [9]. Various state and federal agencies, including the drug enforcement agencies, have been monitoring narcotic prescription patterns [10].

Due to the limited data on hospitalizations for OUD without opioid overdose, detoxification or rehabilitation services, we aimed to examine hospitalizations related to this clinical problem. We examined time-trends in the OUD hospitalizations and the associated healthcare utilization and mortality, and assessed the factors associated with healthcare utilization and mortality during the OUD-associated hospitalizations.

## Materials and methods

### Data source

We included discharges from the Healthcare Cost and Utilization Project's (HCUP) National Inpatient Sample (NIS) from 1998 to 2016. The NIS is a 20% stratified sample of hospital discharges, designed for creating national estimates of all hospitalizations in the U.S. The NIS changed design in 2012 from a 20% sample of hospitals to a 20% sample of discharges from hospitals. We used the recommended trend weights from the HCUP documentation to allow analyses across multiple years. The University of Alabama at Birmingham's Institutional Review Board approved this study (X120207004) and waived the need for informed consent for this database study since these national data are de-identified. All investigations were conducted in conformity with ethical principles of research.

### Study cohort

We identified hospitalizations for OUD based on the presence of any of the following International Classification of Diseases, Ninth Revision, Clinical Modification (ICD-9-CM) or Tenth Revision, Clinical Modification (ICD-10-CM) diagnostic codes for opioid dependence, abuse, or poisoning in the primary diagnosis position, as by the Agency for Healthcare Research and Quality [11]: ICD-9-CM: 304.0x, 304.7x, 305.5x, 965.0x, E850.0, E935.0, F111.xxx, F112.xxx,

T40.1X1x-4x,T40.2X1x-4x, or T40.3X1x-4x. We excluded hospitalizations with ICD-9-CM diagnostic or procedure codes corresponding to drug/alcohol counseling or rehabilitation/ detoxification including diagnostic codes 304.03, 304.73, 305.53, F11.11xx, or F11.21xx and procedure codes 94.45, 94.64–94.69, HZ2xxxx-3xxxx, HZ4xxxx, HZ5xxxx-6xxxx, HZ81xxx-82xxx, HZ84xxx-86xxx, HZ88xxx-89xxx, HZ91xxx-92xxx, HZ94xxx-96xxx, HZ98xxx-99xxx. This approach has been previously used by Peterson et al. [7]

## Outcomes

Study outcomes were index hospitalization healthcare utilization and inpatient mortality. We assessed the length of hospital stay (above/below median), the total hospital charges in U.S. dollars (above/below median for each year) and the discharge disposition, i.e., to home vs. a rehabilitation facility, which included short- or long-term care hospital, skilled nursing facility, intermediate care facility, or a certified nursing facility. We also assessed the inpatient mortality during the index hospitalization.

We assessed several important covariate and potential confounders including socio-demographics (age, sex, race/ethnicity, income [in quartiles]), comorbidity (Deyo-Charlson comorbidity index, a validated measure that included 17 comorbidities, based on the presence of ICD-9-CM codes [12], categorized as 0, 1 and ≥2), insurance payer (Medicare, Medicaid, private, self-pay or other), and hospital characteristics. We categorized hospital location/teaching status as rural, urban non-teaching or urban teaching hospital; hospital bed size as small, medium or large, using the NIS cut-offs that vary by the year; and hospital region as Northeast, Midwest, South, and West.

## Statistical analyses

We assessed summary statistics for the study cohort. We examined the time-trends by examining the rates of hospitalization for OUD as the primary diagnosis from 1998 to 2016 per 100,000 NIS claims. Inpatient mortality rates were similarly assessed over time for those hospitalized with OUD per 100,000 NIS claims and per 100,000 OUD claims.

We examined healthcare utilization outcomes over time. We performed separate multivariable-adjusted logistic regression analyses to assess the factors associated with each OUD hospitalization-related healthcare utilization outcome, i.e., the total hospital charges above/below the median, discharge to a rehabilitation facility vs. home, the length of hospital stay above/ below the median and inpatient mortality. Models included all covariates and potential confounders of interest described in the section above. We calculated the odds ratios (OR) and 95% confidence intervals (CI).

## Patient and public involvement

There was no direct patient involvement in the development of the study question or the execution of the study.

## Results

### Study cohort characteristics

For the study period from 1998 to 2016, we estimated a total of 781,767 OUD hospitalizations. The mean age was 43.7 years (standard error, 0.1), 52% were male, 67% White, and about a quarter each had Medicare, Medicaid or private insurance payer (Table 1). Fifty-four percent of people admitted for OUD were 45 years or younger. The majority (60%) were relatively healthy with a Deyo-Charlson score of zero. OUD hospitalizations were the highest in the

**Table 1. Characteristics of people with opioid use disorder (OUD)\* hospitalizations in the U.S. from 1998–2016.**

| N (%), unless specified otherwise | Primary OUD-hospitalizations N, projected\*\* = 781,767 |
|---|---|
| **Age, Mean (SE); median** | 43.7 (0.10); 42.7 |
| **Age category, in years** | |
| <34 | 248,077 (31.74%) |
| 34–45 | 177,702 (22.74%) |
| >45–55 | 167,100 (21.38%) |
| >55 | 188,645 (24.14%) |
| **Sex** | |
| Male | 407,778 (52.20%) |
| Female | 373,377 (47.80%) |
| **Race** | |
| White | 520,536 (66.59%) |
| Black | 65,868 (8.43%) |
| Hispanic | 48,157 (6.16%) |
| Other/missing | 147,149 (18.82%) |
| **Deyo-Charlson Score** | |
| 0 | 471,282 (60.28%) |
| 1 | 164,829 (21.08%) |
| ≥2 | 145,656 (18.63%) |
| **Hospital Location/Teaching** | |
| Rural | 88,052 (11.55%) |
| Urban nonteaching | 315,210 (41.35%) |
| Urban teaching | 359,059 (47.10%) |
| **Insurance** | |
| Medicaid | 222,798 (28.58%) |
| Medicare | 212,004 (27.19%) |
| Other | 42,709 (5.48%) |
| Private | 179,884 (23.07%) |
| Self | 122,215 (15.68%) |
| **Income Category** | |
| First quartile | 225,968 (29.83%) |
| Second quartile | 202,685 (26.76%) |
| Third quartile | 178,671 (23.59%) |
| Fourth quartile | 150,090 (19.82%) |
| **Hospital Bed size** | |
| Small | 103,493 (13.58%) |
| Medium | 210,779 (27.65%) |
| Large | 448,050 (58.77%) |
| **Hospital Region** | |
| Northeast | 161,462 (21.10%) |
| Midwest | 182,787 (23.88%) |
| South | 277,347 (36.24%) |
| West | 143,745 (18.78%) |
| Outcomes | |
| **Total Hospital Charge, Mean (SE); median, U.S. $** | 23,876 (314); 12,196 |
| **Discharge Status** | |
| Inpatient | 161,826 (23.66%) |
| Home | 522,243 (76.34%) |

*(Continued)*

**Table 1.** (Continued)

| N (%), unless specified otherwise | Primary OUD-hospitalizations N, projected** = 781,767 |
|---|---|
| **Length of Hospital Stay, Mean (SE); median** | 3.6 (0.02); 1.9 |
| **Length of Hospital Stay category, days***** | |
| ≤3 | 571,442 (73.10%) |
| >3 | 210,325 (26.90%) |
| **Died during hospitalization** | 18,394 (2.36%) |
| Discharge Against Medical Advice | 77,323 (9.89%) |

*Opioid drug abuse hospitalizations included those with primary diagnostic code of the following: 304.0x, 304.7x, 305.5x, 965.0x, E850.0, E935.0, F111.xxx, F112.xxx, T40.1X1x-4x,T40.2X1x-4x, or T40.3X1x-4x We excluded hospitalizations with ICD-9-CM diagnostic or procedure codes corresponding to drug/alcohol counseling and rehabilitation/detoxification including diagnostic codes 304.03, 304.73, 305.53, F11.11xx, or F11.21xx and procedure codes 94.45, 94.64–94.69, HZ2xxxx-3xxxx, HZ4xxxx, HZ5xxxx-6xxxx, HZ81xxx-82xxx, HZ84xxx-86xxx, HZ88xxx-89xxx, HZ91xxx-92xxx, HZ94xxx-96xxx, HZ98xxx-99xxx.

** Based on N, actual = 161,056

***The median hospital stay for all NIS hospitalizations was 3 days, which was used to categorize this variable

lowest income classes; 30% in the first quartile and 27% in the second income quartile with the other quartiles near 20% each. We found that 2.4% of people hospitalized primarily for OUD died during hospitalization and 9.9% left against medical advice (Table 1).

## Characteristics of OUD-hospitalizations and outcomes by region

We found that compared to the Northeast, people with OUD-hospitalizations in the other 3 U.S. regions were more likely to be older, female, have Deyo-Charlson comorbidity index score ≥2, have Medicare, be admitted to a hospital with large bed size; and less likely to be White, have Medicaid, be in the highest income quartile, be admitted to urban, teaching hospital (Table 2).

Compared to the Northeast, we found that a slightly lower proportion of OUD hospitalizations in the other 3 U.S. regions had discharge to non-home settings, had hospital length of stay >3 days or left against medical advice (all with p-value <0.001; Table 2). Mean hospital stay was longest in the Northeast; mean hospital charges were the highest in the West followed by Northeast. Differences in in-hospital mortality were also statistically significant, but small in magnitude.

## Outcomes of opioid use disorder hospitalizations by age, sex and race

We noted the people with older age >55 and females with OUD-hospitalization were significantly more likely than younger people and males to be discharged to non-home settings, have hospital charges higher than the median, or hospital stay >3 days (S1 Table). Whites were more likely to be discharged to non-home settings compared to all other race/ethnicities. Differences in mortality by age, sex and race were small.

## Time-trends in opioid use disorder hospitalization and associated mortality and healthcare utilization

OUD hospitalizations were 59.8 per 100,000 of all NIS hospitalizations in the U.S. with any diagnosis in 1998–2000, which increased steadily over the study period to 190.7 per 100,000

**Table 2. OUD-hospitalization characteristics by U.S. hospital region.**

| N (%), unless specified otherwise | Northeast N = 161,462 (21.10%) | Midwest N = 182,787 (23.88%) | South N = 277,347 (36.24%) | West N = 143,745 (18.78%) |
|---|---|---|---|---|
| **Age category, in years*** | | | | |
| <34 | 57,795 (35.80%) | 60,776 (33.25%) | 88,103 (31.77%) | 35,534 (24.75%) |
| 34–45 | 41,284 (25.75%) | 45,645 (24.97%) | 58,727 (21.18%) | 28,298 (19.71%) |
| >45–55 | 32,326 (20.02%) | 37,932 (20.75%) | 60,897 (21.96%) | 32,793 (22.84%) |
| >55 | 30,046 (18.61%) | 38,424 (21.02%) | 69,564 (25.09%) | 46,953 (32.79%) |
| **Sex*** | | | | |
| Male | 97,330 (60.30%) | 95,865 (52.45%) | 136,533 (49.24%) | 69,581 (48.57%) |
| Female | 64,087 (39.70%) | 86,912 (47.55%) | 140,751 (50.76%) | 73,679 (51.43%) |
| **Race*** | | | | |
| White | 113,563 (70.34%) | 99,314 (54.33%) | 204,933 (73.89%) | 94,260 (65.59%) |
| Black | 19,306 (11.96%) | 16,676 (9.12%) | 22,900 (8.26%) | 6,471 (4.50%) |
| Hispanic | 17,193 (10.65%) | 2,290 (1.25%) | 12,583 (4.54%) | 15,444 (10.75%) |
| Other/missing | 11,396 (7.06%) | 64,502 (35.29%) | 36,931 (13.32%) | 27,545 (19.17%) |
| **Deyo-Charlson Score*** | | | | |
| 0 | 103,460 (64.08%) | 110,104 (60.24%) | 165,652 (59.73%) | 80,599 (56.07%) |
| 1 | 32,067 (19.86%) | 40,668 (22.25%) | 58,226 (20.99%) | 30,810 (21.43%) |
| ≥2 | 25,935 (16.06%) | 32,015 (17.51%) | 53,469 (19.28%) | 32,336 (22.50%) |
| **Hospital Location/Teaching*** | | | | |
| Rural | 11,938 (7.39%) | 21,216 (11.73%) | 42,343 (15.30%) | 12,555 (8.76%) |
| Urban nonteaching | 46,363 (28.71%)) | 68,256 (37.73%) | 122,488 (44.26%) | 78,103 (54.52%) |
| Urban teaching | 103,161 (63.89%) | 91,414 (50.54%) | 111,896 (40.44%) | 52,590 (36.71%) |
| **Insurance*** | | | | |
| Medicaid | 1,347,595 (52.11%) | 668,596 (41.73%) | 682,726 (30.95%) | 455,310 (32.37%) |
| Medicare | 389,372 (15.06%) | 324,596 (20.26%) | 510,635 (23.15%) | 366,844 (26.00%) |
| Other | 83,110 (3.21%) | 88,816 (5.54%) | 159,076 (7.21%) | 117,542 (8.33%) |
| Private | 430,734 (16.65%) | 339,809 (21.21%) | 422,857 (19.17%) | 337,234 (23.90%) |
| Self | 335,482 (12.97%) | 180,332 (11.26%) | 430,700 (19.52%) | 134,087 (9.50%) |
| **Income Category*** | | | | |
| First quartile | 34,756 (22.71%) | 56,282 (31.30%) | 99,603 (36.93%) | 27,866 (20.20%) |
| Second quartile | 32,055 (20.95%) | 53,517 (29.57%) | 78,213 (29.00%) | 34,244 (24.82%) |
| Third quartile | 38,096 (24.89%) | 41,539 (22.95%) | 56,275 (20.87%) | 40,108 (29.07%) |
| Fourth quartile | 48,125 (31.45%) | 29,658 (16.39%) | 35,592 (13.20%) | 35,765 (25.92%) |
| **Hospital Bed size*** | | | | |
| Small | 28,650 (17.74%) | 24,854 (13.74%) | 34,092 (12.32%) | 15,897 (11.10%) |
| Medium | 48,136 (29.81%) | 43,233 (23.90%) | 81,264 (29.37%) | 38,146 (26.63%) |
| Large | 84,676 (52.44%) | 112,799 (62.36%) | 161,370 (58.31%) | 89,205 (62.27%) |
| **Outcomes** | | | | |
| **Discharge Status*** | | | | |
| Inpatient | 31,564 (24.24%) | 35,809 (22.27%) | 61,265 (24.56%) | 29,761 (22.99%) |
| Home | 98,658 (75.76%) | 125,022 (77.74%) | 188,208 (75.44%) | 99,695 (77.01%) |
| **Length of Hospital Stay category, days*** | | | | |
| ≤3 | 103,047 (63.82%) | 132,027 (72.73%) | 183,251 (66.07%) | 98,878 (68.79%) |
| >3 | 58,415 (36.18%) | 50,760 (27.77%) | 94,096 (33.93%) | 44,867 (31.21%) |
| **Length of Hospital Stay, Mean (SE)*; median** | 4.1 (0.07); 2.0 | 3.2 (0.05); 1.8 | 3.7 (0.03); 1.9 | 3.6 (0.04); 1.8 |
| **Total Hospital Charges, Mean (SE)*; median** | 24,463 (619); 11,418 | 17,828 (711); 9,714 | 23,134 (275); 12,361 | 34,403 (529); 18,852 |
| **Died during hospitalization*** | 4,083 (2.54%) | 3,986 (2.18%) | 6,368 (2.30%) | 3,857 (2.69%) |

*(Continued)*

**Table 2.** (Continued)

| N (%), unless specified otherwise | Northeast N = 161,462 (21.10%) | Midwest N = 182,787 (23.88%) | South N = 277,347 (36.24%) | West N = 143,745 (18.78%) |
|---|---|---|---|---|
| Discharge Against Medical Advice* | 26,534 (16.43%) | 17,651 (9.66%) | 21,246 (7.66%) | 9,687 (6.74%) |

Total N, projected for OUD-hospitalization = 781,767

*Statistically significantly with a p-value <0.001

NIS hospitalizations in 2015–16, i.e. a 219% increase, leading to a rate increase of 3.2-fold (Table 3; Fig 1). The mortality rate for OUD hospitalization was 1.2 per 100,000 NIS hospitalizations in the U.S. in 1998–2000 that increased 5-times to 5.9 per 100,000 NIS hospitalizations in 2015–16 (Table 3).

Among the OUD hospitalizations, the mortality rate increased from 19.8 per 1,000 OUD hospitalizations in 1998–2000 to 30.9 per 1,000 OUD hospitalizations in 2015–16 (Table 3). In comparison, mortality rate decreased for non-OUD hospitalizations over the same period from 23.8 to 19 per 1,000 non-OUD hospitalizations (Fig 2). Time-related increase in OUD hospitalizations and associated mortality was seen in all age groups, both sexes and in both white and non-white race/ethnicity (**data available on request**).

Time-trends in OUD hospitalization associated healthcare utilization showed an increase over the study period, with the mean (median) total hospital charges increased from $8,261 ($4,339) to $32,792 ($18,244; Table 3). In contrast, we saw little change in mean (median) length of hospital stay from 3.2 days (1.6) to 3.9 days (2.2) over the study period and no change in the proportion discharged to home, i.e., 80% in 1998–2000 versus 80% in 2015–16 (Table 4).

**Table 3. Time-trends in OUD hospitalization and mortality rates from 1998 to 2016 and the comparative non-OUD mortality rates.**

| | Total NIS claims | OUD claims | OUD deaths | OUD claims Per 100K total NIS claims | OUD Death rate Per 100K NIS claims | OUD Death rate per 1k primary OUD claims | Comparative Death rate per 1k Non-OUD claims |
|---|---|---|---|---|---|---|---|
| 1998–2000 | 103,665,051 | 62,010 | 1,226 | 59.82 | 1.18 | 19.77 | 23.79 |
| 2001–2002 | 72,617,381 | 53,176 | 1,002 | 73.23 | 1.38 | 18.84 | 22.18 |
| 2003–2004 | 74,571,583 | 63,853 | 1,228 | 85.63 | 1.65 | 19.23 | 20.88 |
| 2005–2006 | 75,919,595 | 66,923 | 1,350 | 88.15 | 1.78 | 20.17 | 19.73 |
| 2007–2008 | 76,366,797 | 78,541 | 1,524 | 102.85 | 2.00 | 19.40 | 19.07 |
| 2009–2010 | 75,086,597 | 97,611 | 2,149 | 130.00 | 2.86 | 22.02 | 18.24 |
| 2011–2012 | 73,447,261 | 112,428 | 2,445 | 153.07 | 3.33 | 21.75 | 18.07 |
| 2013–2014 | 70,956,610 | 110,985 | 3,255 | 156.41 | 4.59 | 29.33 | 18.93 |
| 2015–2016 | 71,445,363 | 136,240 | 4,215 | 190.69 | 5.90 | 30.94 | 19.05 |

All rates are expressed per 100k or per 1k claims or hospitalizations

The last column represents the death rate in all NIS hospitalizations except OUD hospitalizations.

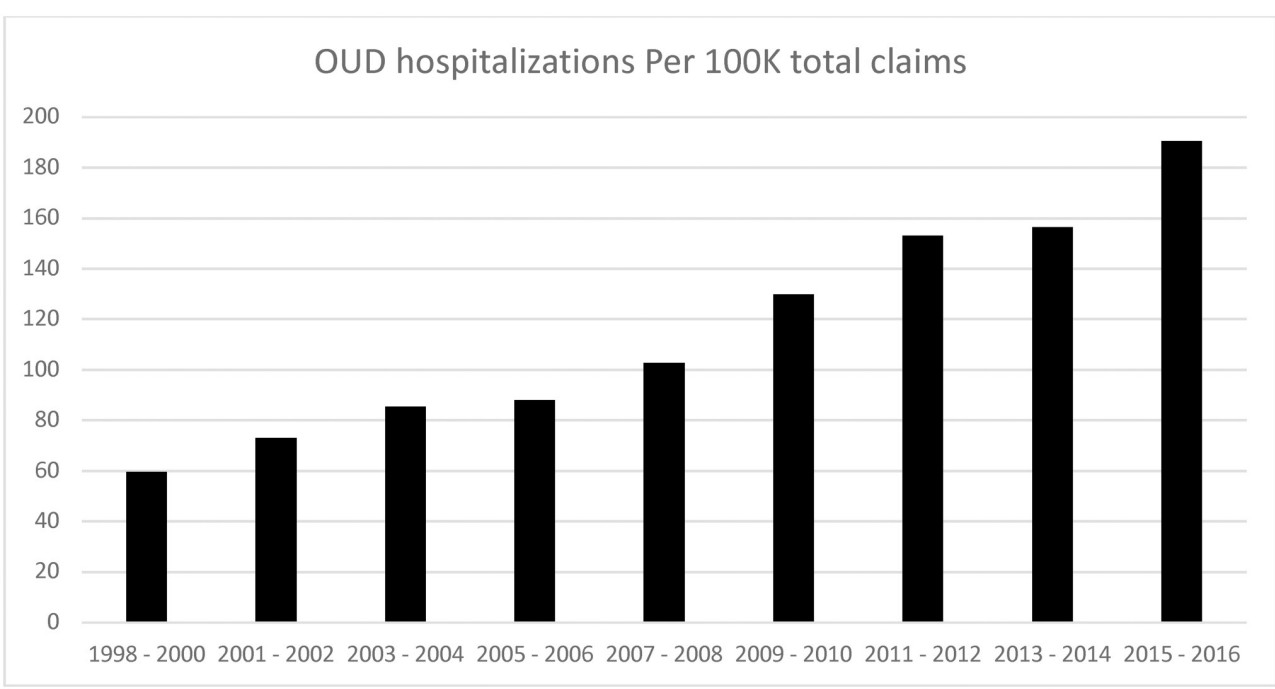

**Fig 1. Time-trend in OUD hospitalization rate per 100,000 NIS claims from 1998 to 2016.** X-axis represents time-periods from 1998–2000 to 2015–16. Y-axis shows primary OUD hospitalization rates per 100,00 NIS claims.

### Multivariable-adjusted predictors of healthcare utilization and inpatient mortality in people admitted with opioid use disorder

In the multivariable-adjusted analysis, compared to age <34 years, older age was associated with a higher risk of hospital charges above the median and the length of hospital stay >3 days and a slightly higher risk of discharge to a rehabilitation facility (Table 5). Higher Deyo-Charlson score was associated with higher hospital charges, a longer length of hospital stay, and higher inpatient mortality (Table 5). Women had 26% higher odds and Blacks 31% lower odds of discharge to a rehabilitation facility, compared to men and Whites, respectively.

We found that women had 25% lower odds of inpatient mortality than men, blacks had 33% lower odds of mortality than whites and older age was associated with higher inpatient mortality. The models for hospital charges, length of stay, and inpatient mortality also showed better outcomes for rural hospitals compared with both urban teaching and urban non-teaching hospitals. Compared to the hospitals in the Northeast U.S., those in the Midwest and the South had lower hospital charges, shorter length of stay, and lower odds of discharge to non-home settings. Lower income was associated with lower hospital charges and lower odds of discharge to non-home settings (Table 5).

### Discussion

We performed a longitudinal study of OUD hospitalizations over a 19-year period from 1998 to 2016, the most recent year of publicly available NIS data. We examined the time-trends in OUD hospitalizations and associated healthcare utilization outcomes and mortality, and their predictors. Our multivariable-adjusted models identified several factors independently associated with each healthcare utilization and in-hospital mortality, while all the other factors shown were adjusted for in the analyses. Several findings of this study merit further discussion.

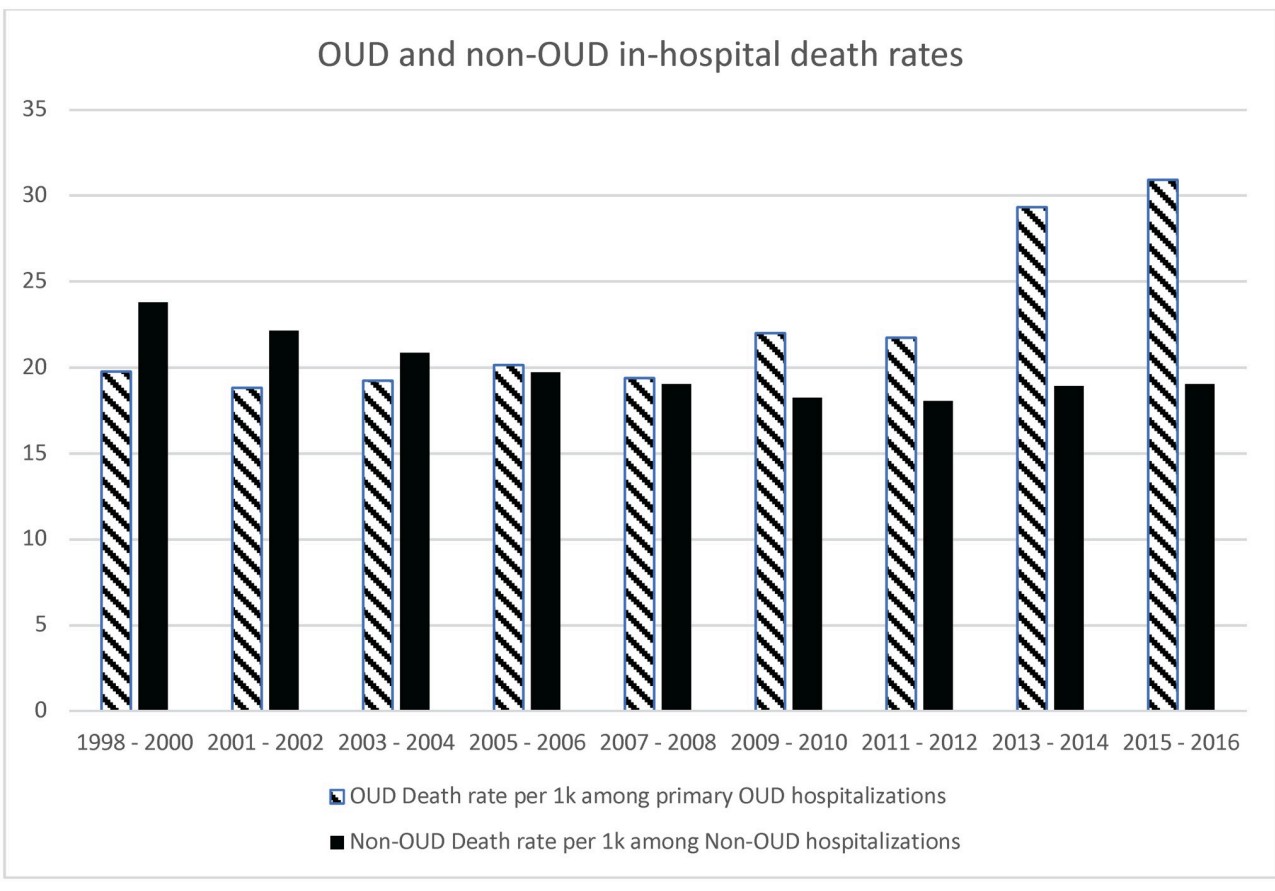

**Fig 2. Comparison of OUD vs. non-OUD death rates over the study period from 1998 to 2016.** X-axis represents time-periods from 1998–2000 to 2015–16. Y-axis shows the in-hospital death rates per 1k among primary OUD hospitalizations (hashed bars) and among all NIS hospitalizations except OUD (solid bars).

The OUD hospitalizations in the U.S. increased steadily from 62,010 in 1998–2000 to 136,240 in 2015–2016, the most recent period with available data. We noted a 219% increase in OUD hospitalizations, compared to the baseline from 1998–2000. The continued rise in OUD hospitalizations in the U.S. is of concern. State and federal agencies have implemented

**Table 4. Time-trends in healthcare utilization outcomes for OUD Hospitalizations from 1998 to 2016.**

|  | Total hospital charges, US $ | Discharged home | Length of Hospital Stay, days |
|---|---|---|---|
|  | Mean (SE); median | N (%) | Mean (SE); median |
| 1998–2000 | 8,261 (601); 4,339 | 39,726 (80%) | 3.2 (0.13); 1.6 |
| 2001–2002 | 11,101 (729); 5,676 | 33,074 (75%) | 3.4 (0.07); 1.8 |
| 2003–2004 | 13,830 (1,003); 7,064 | 41,975 (76%) | 3.4 (0.12); 1.7 |
| 2005–2006 | 17,756 (557); 9,573 | 44,941 (75%) | 3.6 (0.07); 1.8 |
| 2007–2008 | 22,767 (563); 11,912 | 52,091 (74%) | 3.7 (0.08); 1.8 |
| 2009–2010 | 24,844 (648); 13,387 | 65,752 (75%) | 3.7 (0.06); 1.9 |
| 2011–2012 | 28,210 (573); 15,634 | 75,989 (75%) | 3.7 (0.05); 1.9 |
| 2013–2014 | 32,666 (477); 18,188 | 73,985 (75%) | 3.8 (0.04); 1.9 |
| 2015–2016 | 32,792 (586); 18,244 | 94,710 (80%) | 3.9 (0.05); 2.2 |

SE, standard error; US $, US dollar

**Table 5. Predictors of healthcare utilization for people with an OUD hospitalization in the U.S.**

| | Hospital charge above the median[*] | Length of hospital stay > 3 days | Discharge to inpatient facility | In-hospital Mortality |
|---|---|---|---|---|
| Age category | | | | |
| <34 years | Ref | Ref | Ref | Ref |
| 34–45 years | **1.24 (1.20, 1.28)** | **1.06 (1.02, 1.09)** | **1.05 (1.01, 1.09)** | 0.95 (0.86, 1.04) |
| >45–55 years | **1.61 (1.55, 1.66)** | **1.21 (1.17, 1.26)** | 1.02 (0.98, 1.06) | **0.82 (0.74, 0.91)** |
| >55 years | **1.72 (1.66, 1.79)** | **1.41 (1.36, 1.47)** | **1.16 (1.11, 1.21)** | **0.74 (0.66, 0.83)** |
| Sex | | | | |
| Male | Ref | Ref | Ref | Ref |
| Female | 0.97 (0.95, 0.99) | 1.01 (0.99, 1.04) | **1.26 (1.23, 1.30)** | **0.75 (0.70, 0.81)** |
| Race | | | | |
| White | Ref | Ref | Ref | Ref |
| Black | **0.87 (0.84, 0.91)** | **0.84 (0.81, 0.88)** | **0.69 (0.65, 0.73)** | **0.67 (0.59, 0.77)** |
| Hispanic | **1.13 (1.07, 1.18)** | **0.91 (0.86, 0.95)** | **0.83 (0.78, 0.88)** | **0.76 (0.66, 0.88)** |
| Other/missing | **0.84 (0.82, 0.87)** | **0.77 (0.74, 0.79)** | **0.88 (0.85, 0.91)** | **0.90 (0.82, 0.99)** |
| Deyo-Charlson comorbidity Score | | | | |
| 0 | Ref | Ref | Ref | Ref |
| 1 | **1.57 (1.53, 1.62)** | **1.35 (1.31, 1.39)** | **0.92 (0.89, 0.95)** | **1.84 (1.69, 2.02)** |
| ≥2 | **2.16 (2.09, 2.23)** | **1.91 (1.84, 1.97)** | **1.05 (1.01, 1.09)** | **2.46 (2.23, 2.72)** |
| Insurance | | | | |
| Private | Ref | Ref | Ref | Ref |
| Medicaid | 0.97 (0.94, 1.00) | **1.06 (1.03, 1.10)** | **0.82 (0.79, 0.85)** | **1.17 (1.06, 1.29)** |
| Medicare | **1.15 (1.11, 1.19)** | **1.11 (1.07, 1.15)** | **1.20 (1.16, 1.25)** | **0.79 (0.71, 0.88)** |
| Other | **1.08 (1.03, 1.14)** | 1.06 (1.00, 1.12) | **0.89 (0.84, 0.95)** | 0.98 (0.83, 1.15) |
| Self | **1.04 (1.01, 1.08)** | **0.72 (0.69, 0.75)** | **0.75 (0.72, 0.79)** | **1.24 (1.11, 1.38)** |
| Income category | | | | |
| First quartile | **0.86 (0.83, 0.89)** | **1.06 (1.02, 1.10)** | **0.79 (0.76, 0.83)** | 0.96 (0.87, 1.06) |
| Second quartile | **0.86 (0.83, 0.89)** | 1.00 (0.96, 1.03) | **0.88 (0.85, 0.92)** | 0.97 (0.88, 1.07) |
| Third quartile | **0.93 (0.90, 0.96)** | 1.02 (0.98, 1.06) | **0.92 (0.89, 0.96)** | 0.98 (0.89, 1.08) |
| Fourth quartile | Ref | Ref | Ref | Ref |
| Hospital region | | | | |
| Northeast | Ref | Ref | Ref | Ref |
| Midwest | **0.58 (0.56, 0.60)** | **0.71 (0.69, 0.74)** | **0.88 (0.85, 0.92)** | 0.91 (0.82, 1.01) |
| South | **0.90 (0.88, 0.93)** | **0.93 (0.90, 0.97)** | **0.95 (0.91, 0.98)** | 1.00 (0.91, 1.11) |
| West | **1.66 (1.60, 1.72)** | **0.75 (0.72, 0.77)** | **0.82 (0.78, 0.85)** | **1.16 (1.04, 1.29)** |
| Hospital teaching status | | | | |
| Rural | Ref | Ref | Ref | Ref |
| Urban nonteaching | **2.28 (2.19, 2.37)** | **1.42 (1.36, 1.48)** | 0.98 (0.94, 1.03) | **1.57 (1.36, 1.82)** |
| Urban teaching | **2.42 (2.32, 2.52)** | **1.81 (1.74, 1.89)** | **0.84 (0.80, 0.87)** | **2.17 (1.88, 2.50)** |
| Hospital bed size | | | | |
| Small | Ref | Ref | Ref | Ref |
| Medium | **1.35 (1.30, 1.40)** | **1.08 (1.04, 1.12)** | 1.02 (0.98, 1.07) | **1.25 (1.11, 1.40)** |
| Large | **1.53 (1.48, 1.58)** | **1.24 (1.20, 1.29)** | 1.00 (0.96, 1.04) | **1.19 (1.07, 1.32)** |

[*]Total hospital charge were categorized as above or below the median for each year individually

several policies for OUD and various programs to reduce related morbidity [10,13–16]. This increasing trend in OUD hospitalizations in the U.S. confirms the impact of OUD epidemic on the healthcare system, and describes the magnitude of the problem. These findings are also consistent with an increasing OUD in delivery hospitalizations to 2014 [17].

The death rate for OUD hospitalizations was 1.2 per 100,000 NIS hospitalizations in 1998–2000 that increased 5-times to 5.9 per 100,000 NIS hospitalizations in 2015–16. The increase in the OUD hospitalization mortality rate continued through the most recent study period, 2015–16. There was an increase of 77% between 2011–2012 to 2015–16. This is consistent with national CDC estimates of rapidly increasing OUD-related deaths, noted to be 28,647, 33,091 and 42,249 in 2014, 2015 and 2016, respectively [1,6]. The 47% increase in OUD-related mortality from 2014 to 2016 was alarming [6].

Compared to mortality rate in the general population with hospitalization, OUD-related mortality rates were 0.8 times in 1998–2000, but rose to 1.6 times higher in 2015–2016. This indicates worsening of the mortality outcome in OUD-related hospitalizations over time, relative to all other hospitalizations in the U.S. This might be related to a higher severity of opioid abuse, a reduction in access to care or higher associated psychiatric or medical comorbidity over time. These hypotheses need further examination.

In unadjusted comparisons, we noted the OUD hospitalizations in people with older age >55, females and Northeast U.S. region had higher healthcare utilization; OUD hospitalizations in Northeast were also associated with higher proportion of people leaving against medical advice. Whites had higher rate of discharge to non-home settings after OUD hospitalizations compared to other race/ethnicities. Mortality rates were only slightly different by any of these characteristics. We also found interesting differences in patient characteristics by U.S. region in OUD hospitalizations.

The implementation of effective policy and public health programs in the U.S. has the potential to reverse the trend in OUD hospitalizations in the near future [9,10,13–16,18–20]. Strategies and programs to reduce OUD and improve OUD outcomes are being developed. Examples include a system-wide organizational opioid stewardship program (OSP) that was associated with a reduction in opioid morbidity [21]. A combined implementation of mandated provider review of state-run prescription drug monitoring program and pain clinic laws reduced opioid amounts prescribed by 8% and prescription opioid overdose death rates by 12% [22]. Telemedicine has the potential to improve the provision of evidence-based medication-assisted treatment for OUD [23]. The use of buprenorphine and methadone maintenance treatment after non-fatal opioid overdose reduced all-cause and opioid-related mortality [24]. This finding is supported by a systematic review and meta-analysis of 19 studies with 138,716 people treated with either methadone or buprenorphine for opioid dependence [25]. Thus, effective strategies exist to reduce the OUD-related morbidity and mortality.

We examined important patient/clinical characteristics associated with OUD hospitalization related healthcare utilization and mortality. Older age, White race, a higher Deyo-Charlson score and female sex were each associated with worse healthcare utilization outcomes or mortality related to index OUD hospitalization. A previous CDC analysis of drug overdose deaths (prescription opioids and heroin were the main causes) using the 2013 and 2014 national data found that age-adjusted mortality rates for Whites, Blacks and Hispanics were 19, 10.5 and 6.7 per 100,000 [3]. In a study of OUD-hospitalization mortality, Whites, ages 50–64, Medicare beneficiaries with disabilities, and residents of lower-income areas were noted to have higher odds of opioid/heroin poisoning [26]. These studies provide one potential reason for higher mortality in Whites and are consistent with our observation of an independent association of White race with higher mortality during OUD-hospitalization, adjusted for age, sex, insurance, income, comorbidity, hospital region (rural/urban) and teaching status, location or

bed size. Future studies are needed to assess the other underlying causes for higher mortality in Whites with OUD-hospitalizations.

Our observation of the association of male sex with higher inpatient mortality of OUD-hospitalization extends similar observations in people who underwent elective total joint replacement [27] or with pharmaceutical opioid related overdose deaths [28]. We also noted differences by region and by hospital characteristics in these outcomes, which extend similar findings for opioid and heroin-related overdose hospitalizations [8] to OUD-related hospitalizations.

Our study findings must be interpreted considering study limitations. Misclassification bias is possible, since we used diagnostic codes for the identification of the study cohort and comorbidities. Our observational cohort study design puts this study at the potential risk of residual confounding for the predictors of healthcare utilization and mortality outcomes; we adjusted for multiple covariates and confounders to reduce the risk of confounding bias. We assessed hospital charges, which are usually inflated and do not reflect the actual cost of the hospitalization. Due to the lack of cause of death data in the NIS, we are unable to comment on whether the causes of death changed over time, were attributed to OUD or related disorder (hepatitis C, HIV, endocarditis, valvular disease) or differed by factors significantly associated with higher mortality. Longer-term studies of mortality up to 4 years after OUD hospitalization found that both opioid use and physical comorbidities contributed to mortality [29,30].

Our study has many strengths. We used the U.S. NIS, a national dataset that makes our results generalizable to the general U.S. population. We used two decades of data to examine the time-trends in OUD hospitalization, another study strength.

## Conclusions

In conclusion, we found increasing rates of OUD hospitalizations and OUD mortality rates from 1998 to 2016. These time-trends are concerning, given the alarmingly high rates of associated mortality and no trends of a slow-down or decline. We identified factors associated with healthcare utilization and mortality outcomes for OUD hospitalizations. Future studies need to examine the most effective strategies to reduce OUD hospitalizations and associated mortality and healthcare utilization.

## Supporting information

**S1 Table. OUD-hospitalization outcomes by age, sex and race.**
(DOCX)

## Acknowledgments

Disclaimer: The views, presented in this article are solely the responsibility of the author(s) and do not necessarily represent the views of Department of Veterans Affairs.

## Author Contributions

**Conceptualization:** Jasvinder A. Singh.

**Data curation:** John D. Cleveland.

**Formal analysis:** John D. Cleveland.

**Investigation:** Jasvinder A. Singh.

**Methodology:** Jasvinder A. Singh, John D. Cleveland.

**Project administration:** Jasvinder A. Singh, John D. Cleveland.

**Resources:** Jasvinder A. Singh.

**Software:** Jasvinder A. Singh.

**Supervision:** Jasvinder A. Singh.

**Validation:** Jasvinder A. Singh.

**Writing – original draft:** Jasvinder A. Singh.

**Writing – review & editing:** Jasvinder A. Singh, John D. Cleveland.

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
