## [Decision Letter · Decision Letter 0]

20 Nov 2019

PONE-D-19-24166

National U.S. Time-trends in Opioid Use Disorder Hospitalizations and Associated Healthcare Utilization and Mortality

PLOS ONE

Dear Dr Singh,

Thank you for submitting your manuscript to PLOS ONE. After careful consideration, we feel that it has merit but does not fully meet PLOS ONE’s publication criteria as it currently stands. Therefore, we invite you to submit a revised version of the manuscript that addresses the points raised during the review process.

We would appreciate receiving your revised manuscript by Jan 04 2020 11:59PM. To enhance the reproducibility of your results, we recommend that if applicable you deposit your laboratory protocols in protocols.io, where a protocol can be assigned its own identifier (DOI) such that it can be cited independently in the future. For instructions see: http://journals.plos.org/plosone/s/submission-guidelines#loc-laboratory-protocols

We look forward to receiving your revised manuscript.

Kind regards,

Sandra C. Buttigieg, MD PhD FFPH MSc MBA MMCFD

Academic Editor

PLOS ONE

Journal Requirements:

"I have read the journal's policy and the authors of this manuscript have the following competing interests. JAS has received consultant fees from Crealta/Horizon, Medisys, Fidia, UBM LLC, Medscape, WebMD, Clinical Care options, Clearview healthcare partners, Putnam associates, Spherix, the National Institutes of Health and the American College of Rheumatology.  JAS owns stock options in Amarin pharmaceuticals and Viking therapeutics. JAS is a member of the executive of OMERACT, an organization that develops outcome measures in rheumatology and receives arms-length funding from 36 companies. JAS serves on the FDA Arthritis Advisory Committee. JAS is a member of the Veterans Affairs Rheumatology Field Advisory Committee. JAS is the editor and the Director of the UAB Cochrane Musculoskeletal Group Satellite Center on Network Meta-analysis.  JAS previously served as a member of the following committees: member, the American College of Rheumatology's (ACR) Annual Meeting Planning Committee (AMPC) and Quality of Care Committees, the Chair of the ACR Meet-the-Professor, Workshop and Study Group Subcommittee and the co-chair of the ACR Criteria and Response Criteria subcommittee.  JDC has no conflicts. ".

i) Please confirm that this does not alter your adherence to all PLOS ONE policies on sharing data and materials, by including the following statement: "This does not alter our adherence to  PLOS ONE policies on sharing data and materials.” (as detailed online in our guide for authors http://journals.plos.org/plosone/s/competing-interests).  If there are restrictions on sharing of data and/or materials, please state these. Please note that we cannot proceed with consideration of your article until this information has been declared.

ii) Please include your updated Competing Interests statement in your cover letter; we will change the online submission form on your behalf.

Additional Editor Comments (if provided):

I have read this manuscript with great interest. I advise you to take note of the very detailed comments of both reviewers and respond accordingly. I look forward to your revised manuscript.

Reviewers' comments:

Reviewer's Responses to Questions

**Comments to the Author**

1. Is the manuscript technically sound, and do the data support the conclusions?

Reviewer #1: Yes

Reviewer #2: Yes

2. Has the statistical analysis been performed appropriately and rigorously? 

Reviewer #1: Yes

Reviewer #2: Yes

3. Have the authors made all data underlying the findings in their manuscript fully available?

Reviewer #1: No

Reviewer #2: Yes

4. Is the manuscript presented in an intelligible fashion and written in standard English?

Reviewer #1: Yes

Reviewer #2: Yes

5. Review Comments to the Author

Reviewer #1: The authors used 781,000+ hospital records from a National Institutional sample of hospitalizations for the period 1998 through 2016 to calculate rates of hospitalization by age race and gender for Opioid overdose Hospitalizations and in hospital mortality noting that the rates increase from 59.8 per 100,00 to 190 per 100,000 over the 18 years period.

They indicate that on a national basis this is one of the first and only papers to chronicle this information which makes it unique. They examined healthcare utilization outcomes over time. Can they move this forward by examining some of the regional differences in west, midwest, Northeast and south regions , are these all the same with regard to white and black and by age?

A geospatial map of rates or table would be helpful showing black , white Hospitalization rates as well as a regional difference map as opposed to a multivariate logistic regression which is interesting but does not show important nuances in time , person and place.

Much can be learned in the details of this study, Older white male rates of mortality and hospitalization have been detailed previously so presenting some of the details would prove to be important.

A few questions, they indicate older white opioid hospitalizations fared worse that black and younger admits. Please comment on the reasons for this , specifically what were the causes of death among older white admits ; was this just an age phenomenon or was there something specific to region or urban /rural.

Please show stratification by average days of hospitalization by age and gender and race.

OUD and old age can be linked to end stage cancer diagnoses, please provide the information of end of life related causes of death versus others.

How much of the co morbidity was opioid and drug related, aka, Hepatitis C, HIV and myocardial endocarditis and valvular disease?

ALso the logistic regressions should be given more detail with tables shown. What was the median length of stay hospitalization for those under 55 and over 55?

This data set has the ability to show important ground breaking information .

Reviewer #2: The authors conducted an observational study form National Inpatient Sample, and showed an increase in OUD hospitalizations and associated inpatient mortality. Age, sex, race, and Deyo-Charlson comorbidity index are independent factors of healthcare utilization and mortality related to index OUD hospitalization.

Is there any data to support these independent factors associated with healthcare utilization and mortality related to OUD hospitalization?

Some comorbidity indices have been developed to measure and weigh the overall burden of comorbidities, for example, Charlson comorbidity index(CCI), Deyo CCI, Romano CCI. Is there any special reason for choosing Deyo CCI in this article.

Minor point: Table 4: The form after “Rural” is empty. Should it be “Ref”?

6. PLOS authors have the option to publish the peer review history of their article (what does this mean?). If published, this will include your full peer review and any attached files.

Reviewer #1: No

Reviewer #2: No

---

## [Author Response · Author response to Decision Letter 0]

27 Nov 2019

We thank the reviewers for their comments. Following are our point-by-point responses to their comments. 

Reviewers' comments:

Reviewer's Responses to Questions

Comments to the Author

1. Is the manuscript technically sound, and do the data support the conclusions?

Reviewer #1: Yes

Reviewer #2: Yes

Response: Thank you. 

2. Has the statistical analysis been performed appropriately and rigorously?

Reviewer #1: Yes

Reviewer #2: Yes 

Response: Thank you.

3. Have the authors made all data underlying the findings in their manuscript fully available?

Reviewer #1: No

Reviewer #2: Yes

Response: Thank you. AHRQ data have to be obtained directly from AHRQ. AHRQ does not allow individual researcher to disseminate the data. These de-identified data are publically available. We had included this statemnet in the data sharing section. 

4. Is the manuscript presented in an intelligible fashion and written in standard English?

Reviewer #1: Yes

Reviewer #2: Yes 

Response: Thank you

5. Review Comments to the Author

Reviewer #1: The authors used 781,000+ hospital records from a National Institutional sample of hospitalizations for the period 1998 through 2016 to calculate rates of hospitalization by age race and gender for Opioid overdose Hospitalizations and in hospital mortality noting that the rates increase from 59.8 per 100,00 to 190 per 100,000 over the 18 years period.

They indicate that on a national basis this is one of the first and only papers to chronicle this information which makes it unique. They examined healthcare utilization outcomes over time. Can they move this forward by examining some of the regional differences in west, midwest, Northeast and south regions , are these all the same with regard to white and black and by age?

Response: We have provided this as a new table as suggested, and have these results to results and discussion sections.

Table 2. OUD-hospitalization characteristics by U.S. hospital region

N (%), unless specified otherwise Northeast 

N=161,462 (21.10%)

 Midwest

N=182,787 (23.88%) South N=277,347 (36.24%) West 

N=143,745 (18.78%)

Age category, in years* 

 <34 57,795 (35.80%) 60,776 (33.25%) 88,103 (31.77%) 35,534(24.75%)

 34 - 45 41,284 (25.75%) 45,645 (24.97%) 58,727 (21.18%) 28,298 (19.71%)

 >45 - 55 32,326 (20.02%) 37,932 (20.75%) 60,897 (21.96%) 32,793 (22.84%)

 >55 30,046 (18.61%) 38,424 (21.02%) 69,564 (25.09%) 46,953 (32.79%)

Sex* 

 Male 97,330 (60.30%) 95,865 (52.45%) 136,533 (49.24%) 69,581 (48.57%)

 Female 64,087 (39.70%) 86,912 (47.55%) 140,751 (50.76%) 73,679 (51.43%)

Race* 

 White 113,563 (70.34%) 99,314 (54.33%) 204,933 (73.89%) 94,260 (65.59%)

 Black 19,306 (11.96%) 16,676 (9.12%) 22,900 (8.26%) 6,471 (4.50%)

 Hispanic 17,193 (10.65%) 2,290 (1.25%) 12,583 (4.54%) 15,444 (10.75%)

 Other/missing 11,396 (7.06%) 64,502 (35.29%) 36,931 (13.32%) 27,545 (19.17%)

Deyo-Charlson Score* 

 0 103,460 (64.08%) 110,104 (60.24%) 165,652 (59.73%) 80,599 (56.07%)

 1 32,067 (19.86%) 40,668 (22.25%) 58,226 (20.99%) 30,810 (21.43%)

 ≥2 25,935 (16.06%) 32,015 (17.51%) 53,469 (19.28%) 32,336 (22.50%)

Hospital Location/Teaching* 

 Rural 11,938 (7.39%) 21,216 (11.73%) 42,343 (15.30%) 12,555 (8.76%)

 Urban nonteaching 46,363 (28.71%)) 68,256 (37.73%) 122,488 (44.26%) 78,103 (54.52%)

 Urban teaching 103,161 (63.89%) 91,414 (50.54%) 111,896 (40.44%) 52,590 (36.71%)

Insurance* 

 Medicaid 1,347,595 (52.11%) 668,596 (41.73%) 682,726 (30.95%) 455,310 (32.37%)

 Medicare 389,372 (15.06%) 324,596 (20.26%) 510,635 (23.15%) 366,844 (26.00%)

 Other 83,110 (3.21%) 88,816 (5.54%) 159,076 (7.21%) 117,542 (8.33%)

 Private 430,734 (16.65%) 339,809 (21.21%) 422,857 (19.17%) 337,234 (23.90%)

 Self 335,482 (12.97%) 180,332 (11.26%) 430,700 (19.52%) 134,087 (9.50%)

Income Category* 

 First quartile 34,756 (22.71%) 56,282 (31.30%) 99,603 (36.93%) 27,866 (20.20%)

 Second quartile 32,055 (20.95%) 53,517 (29.57%) 78,213 (29.00%) 34,244 (24.82%)

 Third quartile 38,096 (24.89%) 41,539 (22.95%) 56,275 (20.87%) 40,108 (29.07%)

 Fourth quartile 48,125 (31.45%) 29,658 (16.39%) 35,592 (13.20%) 35,765 (25.92%)

Hospital Bed size* 

 Small 28,650 (17.74%) 24,854 (13.74%) 34,092 (12.32%) 15,897 (11.10%)

 Medium 48,136 (29.81%) 43,233 (23.90%) 81,264 (29.37%) 38,146 (26.63%)

 Large 84,676 (52.44%) 112,799 (62.36%) 161,370 (58.31%) 89,205 (62.27%)

Outcomes

Discharge Status* 

 Inpatient 31,564 (24.24%) 35,809 (22.27%) 61,265 (24.56%) 29,761 (22.99%)

 Home 98,658 (75.76%) 125,022 (77.74%) 188,208 (75.44%) 99,695 (77.01%)

Length of Hospital Stay category, days* 

 ≤3 103,047 (63.82%) 132,027 (72.73%) 183,251 (66.07%) 98,878 (68.79%)

 >3 58,415 (36.18%) 50,760 (27.77%) 94,096 (33.93%) 44,867 (31.21%)

Died during hospitalization* 4,083 (2.54%) 3,986 (2.18%) 6,368 (2.30%) 3,857 (2.69%)

Discharge Against Medical Advice* 26,534 (16.43%) 17,651 (9.66%) 21,246 (7.66%) 9,687 (6.74%)

Total N, projected for OUD-hospitalization = 781,767

*Statistically significantly with a p-value <0.001

Results

“Characteristics of OUD-hospitalizations and outcomes by Region

We found that compared to the Northeast, people with OUD-hospitalizations in the other 3 U.S. regions were more likely to be older, female, have Deyo-Charlson comorbidity index score ≥2, have Medicare, be admitted to a hospital with large bed size; and less likely to be White, have Medicaid, be in the highest income quartile, be admitted to urban, teaching hospital (Table 2). 

Compared to the Northeast, we found that a slightly lower proportion of OUD hospitalizations in the other 3 U.S. regions had discharge to non-home settings, had hospital length of stay >3 days or left against medical advice (all with p-value <0.001; Table 2). Differences in in-hospital mortality were also statistically significant, but small in magnitude.”

Discussion

“In unadjusted comparisons, we noted the OUD hospitalizations in people with older age >55, females and Northeast U.S. region had higher healthcare utilization; OUD hospitalizations in Northeast were also associated with higher proportion of people leaving against medical advice. Whites had higher rate of discharge to non-home settings after OUD hospitalizations compared to other race/ethnicities. Mortality rates were only slightly different by any of these characteristics. We also found interesting differences in patient characteristics by U.S. region in OUD hospitalizations.”

A geospatial map of rates or table would be helpful showing black , white Hospitalization rates as well as a regional difference map as opposed to a multivariate logistic regression which is interesting but does not show important nuances in time , person and place.

Much can be learned in the details of this study, Older white male rates of mortality and hospitalization have been detailed previously so presenting some of the details would prove to be important.

Response: We have provided these data as requested as a new appendix with regards to age, race and sex with time-trends. We have added text to results and discussion. 

“Outcomes of OUD hospitalizations by age, sex and race

We noted the people with older age >55 and females with OUD-hospitalization were significantly more likely than younger people and males to be discharged to non-home settings, have hospital charges higher than the median, or hospital stay >3 days (Appendix 1). Whites were more likely to be discharged to non-home settings compared to all other race/ethnicities. Differences in mortality by age, sex and race were small.

Appendix 1. OUD-hospitalization outcomes by age, sex and race 

N (%), unless specified otherwise % Discharged to non-home settings Total hospital charges

>median Length of hospital stay >3 days Died during hospitalization 

Age category, in years 

 <34 43,787 (21.05%) 73,411 (29.59%) 64,217 (25.89%) 5,963 (2.41%) 

 34 - 45 33,663 (22.38%) 63,307 (35.63%) 51,860 (29.18%) 4,284 (2.42%) 

 >45 - 55 34.309 (23.05%) 76,215 (45.61%) 58,458 (34.98%) 3,903 (2.34%) 

 >55 50,051 (28.33%) 99,568 (52.78%) 78,813 (41.78%) 4,229 (2.24%) 

Sex 

 Male 73,108 (21.10%) 159,035 (39.00%) 127,738 (31.33%) 10,871 (2.67%) 

 Female 88,638 (26.29%) 153,372 (41.08%) 125,518 (33.62%) 7,523 (2.02%) 

Race 

 White 116,236 (25.32%) 215,485 (41.40%) 173,431 (33.32%) 12,770 (2.46%) 

 Black 10,004 (17.57%) 26,722 (40.57%) 22,691 (34.45%) 1,349 (2.05%) 

 Hispanic 7,759 (19.12%) 22,602 (46.93%) 15,801 (32.82%) 1,154 (2.40%) 

 Other/missing 27,822 (17.19%) 47,759 (32.46%) 41,489 (28.20%) 3,116 (2.12%) 

A few questions, they indicate older white opioid hospitalizations fared worse that black and younger admits. Please comment on the reasons for this , specifically what were the causes of death among older white admits ; was this just an age phenomenon or was there something specific to region or urban /rural.

Response: We have more discussion related to this association noted between race and in-hospital mortality. However, due to non-availability of cause of death in this dataset, we are unable to provide further insights into causes of death. Since these analyses were adjusted for age, sex, insurance, income, comorbidity, hospital region (rural/urban) and teaching status, location or bed size, none of the noted difference by race can be attributed to these factors.

“A previous CDC analysis of drug overdose deaths (prescription opioids and heroin were the main causes) using the 2013 and 2014 national data found that age-adjusted mortality rates for Whites, Blacks and Hispanics were 19, 10.5 and 6.7 per 100,000 [3]. This previous observation is consistent with our observation of an independent association of White race with higher mortality during OUD-hospitalization, adjusted for age, sex, insurance, income, comorbidity, hospital region (rural/urban) and teaching status, location or bed size. Future studies are needed to assess the underlying causes for higher mortality for Whites in OUD-hospitalizations.”

Please show stratification by average days of hospitalization by age and gender and race.

Response: Please see the new table and response added in response to an earlier comment by the reviewer. The length of stay >3 days seemed to more in older people, therefore the association of age with LOS was noted in OUD-hospitalization. We noted smaller differences in the length of hospitalization by sex and race. 

OUD and old age can be linked to end stage cancer diagnoses, please provide the information of end of life related causes of death versus others.

Response: We do not have cause of death data in the NIS, therefore we can not definitively assess the cause of OUD-in-hospital death. As can be seen from the table above added per the reviewer request, most of the in-hospital mortality were in the younger people, i.e., 45% of deaths were in people <45 years and 77% of the deaths were in people <55 years. Therefore, it is unlikely that cancer is one of the main reasons for OUD-hospitalization associated death. It is more likely that OUD mortality is related to OUD- and associated complications, including medical complications. 

How much of the co morbidity was opioid and drug related, aka, Hepatitis C, HIV and myocardial endocarditis and valvular disease?

Response: The NIS does not provide the cause of death, so we are unable to ascertain the attribution of death to medical comorbidities listed in the comment vs. the OUD itself. We have added this to the study limitations section. 

“Due to the lack of cause of death data in the NIS, we are unable to comment on whether the causes of death changed over time, were attributed to OUD or related disorder (hepatitis C, HIV, endocarditis, valvular disease) or differed by factors significantly associated with higher mortality.”

ALso the logistic regressions should be given more detail with tables shown. What was the median length of stay hospitalization for those under 55 and over 55?

This data set has the ability to show important ground breaking information .

Response: We have provided additional data as suggested in response to an earlier comment to show data by age, sex and race. This has been added as an appendix/table. In addition, we have added more detail to the results section related to the logistic regression results. 

“In the multivariable-adjusted analysis, compared to age <34 years, older age was associated with a higher risk of hospital charges above the median and the length of hospital stay >3 days and a slightly higher risk of discharge to a rehabilitation facility (Table 4). Higher Deyo-Charlson score was associated with higher hospital charges, a longer length of hospital stay, and higher inpatient mortality (Table 4). Women had 26% higher odds and Blacks 31% lower odds of discharge to a rehabilitation facility, compared to men and Whites, respectively.

We found that women had 25% lower odds of inpatient mortality than men, blacks had 33% lower odds of mortality than whites and older age was associated with higher inpatient mortality. The models for hospital charges, length of stay, and inpatient mortality also showed better outcomes for rural hospitals compared with both urban teaching and urban non-teaching hospitals . Compared to the hospitals in the Northeast U.S., those in the Midwest and the South had lower hospital charges, shorter length of stay, and lower odds of discharge to non-home settings. Lower income was associated with lower hospital charges and lower odds of discharge to non-home settings (Table 4).”

Discussion

“We performed a longitudinal study of OUD hospitalizations over a 19-year period from 1998 to 2016, the most recent year of publicly available NIS data. We examined the time-trends in OUD hospitalizations and associated healthcare utilization outcomes and mortality, and their predictors. Our multivariable-adjusted models identified several factors independently associated with each healthcare utilization and in-hospital mortality, while all the other factors shown were adjusted for in the analyses.”

Reviewer #2: The authors conducted an observational study form National Inpatient Sample, and showed an increase in OUD hospitalizations and associated inpatient mortality. Age, sex, race, and Deyo-Charlson comorbidity index are independent factors of healthcare utilization and mortality related to index OUD hospitalization.

Is there any data to support these independent factors associated with healthcare utilization and mortality related to OUD hospitalization?

Response: We have expanded the discussion related to associations we noted, that support our findings. In general, many findings were from smaller sub-cohorts of people with OUD. 

“We examined important patient/clinical characteristics associated with OUD hospitalization related healthcare utilization and mortality. Older age, White race, a higher Deyo-Charlson score and female sex were each associated with worse healthcare utilization outcomes or mortality related to index OUD hospitalization. A previous CDC analysis of drug overdose deaths (prescription opioids and heroin were the main causes) using the 2013 and 2014 national data found that age-adjusted mortality rates for Whites, Blacks and Hispanics were 19, 10.5 and 6.7 per 100,000 [3]. In a study of OUD-hospitalization mortality, Whites, ages 50–64, Medicare beneficiaries with disabilities, and residents of lower-income areas were noted to have higher odds of opioid/heroin poisoning [26]. These studies provide one potential reason for higher mortality in Whites and are consistent with our observation of an independent association of White race with higher mortality during OUD-hospitalization, adjusted for age, sex, insurance, income, comorbidity, hospital region (rural/urban) and teaching status, location or bed size. Future studies are needed to assess the other underlying causes for higher mortality in Whites with OUD-hospitalizations. 

Our observation of the association of male sex with higher inpatient mortality of OUD-hospitalization extends similar observations in people who underwent elective total joint replacement [27] or with pharmaceutical opioid related overdose deaths [28]. We also noted differences by region and by hospital characteristics in these outcomes, which extend similar findings for opioid and heroin-related overdose hospitalizations [8] to OUD-related hospitalizations.”

Some comorbidity indices have been developed to measure and weigh the overall burden of comorbidities, for example, Charlson comorbidity index(CCI), Deyo CCI, Romano CCI. Is there any special reason for choosing Deyo CCI in this article.

Response: All versions of Charlson index have been validated, we agree whole-heartedly with the reviewer. We chose the Deyo-Charlson version as one of the commonly used versions of this comorbidity index. 

Minor point: Table 4: The form after “Rural” is empty. Should it be “Ref”?

 Response: Thank you for catching this error, we have fixed it. 

6. PLOS authors have the option to publish the peer review history of their article (what does this mean?). If published, this will include your full peer review and any attached files.

Do you want your identity to be public for this peer review? For information about this choice, including consent withdrawal, please see our Privacy Policy.

Reviewer #1: No

Reviewer #2: No

---

## [Decision Letter · Decision Letter 1]

3 Feb 2020

National U.S. Time-trends in Opioid Use Disorder Hospitalizations and Associated Healthcare Utilization and Mortality

PONE-D-19-24166R1

Dear Dr. Singh,

We are pleased to inform you that your manuscript has been judged scientifically suitable for publication and will be formally accepted for publication once it complies with all outstanding technical requirements.

With kind regards,

Sandra C. Buttigieg, MD PhD FFPH MSc MBA MMCFD

Academic Editor

PLOS ONE

Additional Editor Comments (optional):

Reviewers' comments:

Reviewer's Responses to Questions

**Comments to the Author**

1. If the authors have adequately addressed your comments raised in a previous round of review and you feel that this manuscript is now acceptable for publication, you may indicate that here to bypass the “Comments to the Author” section, enter your conflict of interest statement in the “Confidential to Editor” section, and submit your "Accept" recommendation.

Reviewer #1: All comments have been addressed

Reviewer #2: All comments have been addressed

2. Is the manuscript technically sound, and do the data support the conclusions?

Reviewer #1: Yes

Reviewer #2: Yes

3. Has the statistical analysis been performed appropriately and rigorously? 

Reviewer #1: Yes

Reviewer #2: Yes

4. Have the authors made all data underlying the findings in their manuscript fully available?

Reviewer #1: Yes

Reviewer #2: Yes

5. Is the manuscript presented in an intelligible fashion and written in standard English?

Reviewer #1: Yes

Reviewer #2: Yes

6. Review Comments to the Author

Reviewer #1: I have reviewed the revised version and believe the authors have satisfactorily addressed the comments of the reviewers. My only remaining concern is that when considering hospitalization data in a life table way, one would usually remove those individuals who have a primary secondary diagnosis of cancer as often an opioid overuse/abuse

may be due to end stage cancer and pain relief which is considered acceptable in most settings, In other studies this is acknowledged and those individuals are removed prior to the analysis and/or are enumerated in the preliminary methods portion of the paper, I did not see this and so will leave it up to the editors to determine if they wish the authors to further comment on this adjustment.

Reviewer #2: The author's responce is complete, espeically in the part of first reviewer. I think expanded discussion and additional result will make this article meaningful.

7. PLOS authors have the option to publish the peer review history of their article (what does this mean?). If published, this will include your full peer review and any attached files.

Reviewer #1: No

Reviewer #2: Yes: Hsien-Yuan, Chang

---

## [Editor Report · Acceptance letter]

7 Feb 2020

PONE-D-19-24166R1 

National U.S. time-trends in opioid use disorder hospitalizations and associated healthcare utilization and mortality 

Dear Dr. Singh:

I am pleased to inform you that your manuscript has been deemed suitable for publication in PLOS ONE. Congratulations! Your manuscript is now with our production department. 

With kind regards,

on behalf of

Professor Sandra C. Buttigieg 

Academic Editor

PLOS ONE